# The Phytochemical Profile and Biological Activity of *Malva neglecta* Wallr. in Surgically Induced Endometriosis Model in Rats

**DOI:** 10.3390/molecules27227869

**Published:** 2022-11-15

**Authors:** Esra Küpeli Akkol, Büşra Karpuz, Gizem Türkcanoğlu, Fatma Gül Coşgunçelebi, Hakkı Taştan, Michael Aschner, Anurag Khatkar, Eduardo Sobarzo-Sánchez

**Affiliations:** 1Department of Pharmacognosy, Faculty of Pharmacy, Gazi University, 06330 Ankara, Turkey; 2Department of Pharmacognosy, Faculty of Pharmacy, Başkent University, 06810 Ankara, Turkey; 3Department of Biology, Faculty of Science, Gazi University, 06560 Ankara, Turkey; 4Department of Molecular Pharmacology, Albert Einstein College of Medicine, Bronx, NY 10461, USA; 5Faculty of Pharmaceutical Sciences, Maharshi Dayanand University, Rohtak 124001, India; 6Department of Organic Chemistry, Faculty of Pharmacy, University of Santiago de Compostela, 15782 Santiago de Compostela, Spain; 7Facultad de Ciencias de la Salud, Instituto de Investigación y Postgrado, Universidad Central de Chile, Santiago 8330507, Chile

**Keywords:** Malvaceae, *Malva neglecta*, cytokine, endometriosis, polyphenol, quantitative real-time polymerase chain reaction

## Abstract

Leaves and aerial parts of *Malva neglecta* Wallr. have been traditionally used in Anatolia for the treatment of pain, inflammation, hemorrhoids, renal stones, constipation, and infertility. This study investigated the effects of *M. neglecta* leaves in a rat endometriosis model. The dried plant material was extracted with *n*-hexane, ethyl acetate, and methanol, successively. Experimental endometriosis was surgically induced in six-week-old female, non-pregnant, Wistar albino rats by autotransplant of endometrial tissue to the abdominal wall. After twenty-eight days, rats were evaluated for a second laparotomy. Endometrial foci areas were assessed, and intraabdominal adhesions were scored. Rats were divided into five groups as control, *n*-hexane, ethyl acetate, methanol, and aqueous extracts, as well as reference. At the end of the treatment, all rats were sacrificed and endometriotic foci areas and intraabdominal adhesions were re-evaluated and compared with the previous findings. Moreover, peritoneal fluid was collected to detect tumor necrosis factor- α (TNF-α), vascular endothelial growth factor (VEGF), and interleukin-6 (IL-6) levels, and cDNA synthesis, and a quantitative real-time polymerase chain reaction (PCR) test was done. The phytochemical content of the most active extract was determined using High-Performance Liquid Chromatography (HPLC). Both endometrial volume and adhesion score decreased significantly in the group treated with methanol extract. In addition, significant decreases were observed in TNF-α, VEGF, and IL-6 levels in animals administered methanol extract. HPLC results showed that the activity caused by the methanol extract of *M. neglecta* was due to the polyphenols. Taken together, these novel findings indicate that *M. neglecta* may be a promising alternative for the treatment of endometriosis.

## 1. Introduction

Endometriosis is a condition with symptoms such as dysmenorrhea, dyspareunia, pelvic pain, and infertility, caused by the development of endometrial tissue in the rectum, diaphragm, urinary and gastrointestinal system organs, especially the ovaries, outside the uterine cavity [1,2,3]. The establishment and growth of such endometriotic tissue are estrogen-dependent, thus, it is mostly found in women of reproductive age, although the clinical consequences of endometriosis and its management can last well into post-menopause [4,5]. The exact prevalence of endometriosis is unknown, but estimates range from 2 to 10% within the general female population, but up to 50% in infertile women. Thus, it is estimated that currently at least 190 million women and adolescent girls worldwide are affected by the disease during reproductive age, although some women may suffer beyond menopause. Abdominopelvic pain, dysmenorrhea, heavy menstrual bleeding, infertility, dyspareunia/postcoital bleeding, urinary tract symptoms, ovarian cyst, irritable bowel syndrome, pelvic inflammatory disease, or fibrocystic breast disease are risk factors for diagnosis of endometriosis [6]. As in adults, the pathophysiology of endometriosis in adolescents is largely unknown. Endometriosis has been described not only in post-menarcheal girls, possibly resulting in retrograde menstruation, but also in prepubertal but post-thelarche girls, suggesting multifactorial peripubertal etiologies of the disease in the adolescent population [7]. Treatment aims to restore the normal pelvic anatomy, destroy the formed endometrial tissue, relieve existing pelvic pain, and improve the quality of life with infertility treatment. However, despite the treatments, recurrence of the disease is often encountered. Accordingly, it is imperative to find alternative treatments, as the efficacy of contemporary drugs for endometriosis is insufficient and the potential for side effects is high. Medicinal plants have been used traditionally for centuries as potential therapeutics. Although interest in plants decreased with the proliferation of synthetic drugs, the use of plants as treatments has regained interest since the 1990s [8,9].

*Malva neglecta* belongs to the Malvaceae family, locally called “Küçük ebegümeci, çoban yatağı, tolık”, it grows in fields, on roadsides, and in sun-drenched parts of barren places in the main northern and southern regions of Turkey [10]. Various parts of *M. neglecta* are frequently used in folk medicine. In general, the above-ground parts of the plant are preferred and used by infusion with water or as porridge [11]. 

Fruit and aerial parts have been utilized as a traditional remedy in Turkey for the treatment of stomach ache, urinary inflammation, hemorrhoid, cold, rheumatism, sprain, bruising, cough, bronchitis, constipation, and wound healing, as an abortifacient, and for urolithiasis [12,13,14,15]. Decoction prepared from fresh leaves of the plant has also been used to treat dysmenorrhea and infertility [16,17].

Several biological activity tests have been carried out on *M. neglecta*, focusing on its antimicrobial, antioxidant, anti-inflammatory, analgesic, anti-ulcerogenic, wound healing, anticancer, antiemetic, neuroprotective, and antidiabetic activities [18,19,20,21,22,23]. Furthermore, phytochemical studies have characterized the chemical content of different aerial parts of *M. neglecta*, which contain phenolic compounds and acids, amino acids, sugars, and saturated, as well as unsaturated, fatty acids [13,18,24,25,26].

In this study, we investigated the efficacy of different polarity extracts from the leaves of *M. neglecta*, which are frequently used in the traditional treatment of female infertility in our country, internally or externally, in an endometriosis model, to determine the compounds responsible for the efficacy of the active extract.

## 2. Results

### 2.1. Composition of Methanol Extract Prepared from Malva neglecta Leaves

The methanol extract prepared from the leaves of *M. neglecta*, which was determined to be effective in the endometriosis rat model, was used to investigate the compound/s responsible for the effect, using HPLC.

Eighteen polyphenolics commonly found in plant material were characterized and quantified. Peaks co-eluting first were p-hydroxybenzoic acid and coumaroyl hexoside. Hydroxytyrosol of high composition (107.1 mg/g) was followed by coumaroylhexoside (97.2 mg/g). Less than 10 mg/g each Tri-O-galloylquinic acid, epicatechin, ferulic acid, 3-O-caffeoylquinic acid, 5-O-caffeoylquinic acid, 3,4-di-O-caffeoylquinic acid, 3,5-di-O-caffeoylquinic acid and 4,5-di-O-caffeoylquinic acid were determined. The individual phenolic compounds, with their detailed properties, are shown in Table 1 and Figure 1. 

### 2.2. Evaluation of Adhesion Scores

When the intra-abdominal adhesions were classified for the groups, widespread thick omental adhesions (score 3) were detected for all groups before the treatment. In some cases, adhesions were detected between the small intestine and the inner lining of the abdominal wall (score 4). At the end of the experiment, adhesions were observed in the control group animals. On the other hand, no adhesion (score 0) was detected in the reference drug (buserelin acetate) and thin adhesions (score 1) were detected in methanol extract-treated groups. The *n*-hexane extract prepared from the plant showed 9.4% adhesion formation. Although the ethyl acetate extract reduced it by 18.2% and the aqueous extract by 6.5%, these values did not attain statistically significant improvement when compared to the control group. Furthermore, the methanol extract significantly reduced adhesion formation by 57.6% (Table 2). 

### 2.3. Evaluation of Endometriotic Implant Volumes

Post-treatment volumes were found to be significantly decreased, in the reference and methanol extract-treated groups, to 31.63 (*p* < 0.001) and 47.81 mm^3^ (*p* < 0.01), respectively. On the other hand, there was no statistically significant difference between the pre-and post-treatment volumes of the other groups (Figure 2 and Figure 3). 

### 2.4. Evaluation of Cytokine Levels

Significant differences were detected between the peritoneal TNF-α, VEGF, and IL-6 levels before and after treatment in the methanol extract and reference (buserelin acetate) groups when compared to the control group (Table 3). After methanol extract treatment the levels of TNF-α, VEGF, and IL-6 decreased to 4.9 (*p* < 0.05), 12.1 (*p* < 0.05), and 34.1 (*p* < 0.05), respectively. The cytokine levels were extremely and significantly reduced after treatment with buserelin acetate.

### 2.5. TNF-alpha and VEGF Expression by Real-Time PCR

On the other hand, *n*-hexane, ethyl acetate, and the aqueous extracts failed to statistically or significantly decrease TNF-α, VEGF, and IL-6 levels, compared with the control group. The results showed that *M. neglecta* contributed to the treatment by significantly attenuating the increased cytokine levels in endometriosis. 

Real-time PCR was carried out to evaluate the levels of TNF-α and VEGF mRNA expressions. Both VEGF and TNF-α mRNA expressions decreased in the experimental groups. Among the experimental groups, VEGF mRNA expression was found to be the lowest in the buserelin acetate group (* *p* < 0.001). The relative VEGF mRNA expression was also decreased by methanol, aqueous, ethyl acetate, *n*-hexane, and the control groups (Figure 4A). TNF-α mRNA expression decreased in the experimental groups when compared with the control group. Among the experimental groups, TNF-alpha mRNA expression was the lowest in the buserelin acetate group (* *p* < 0.002), followed by decreased expression levels in the methanol, aqueous, ethyl acetate, n-hexane, and control groups (Figure 4B).

### 2.6. Histopathological Analyses

In the histopathological analyses, endometrial glands and inflammatory cell infiltration (CI) were observed in the control groups, and a decrease in these values was detected in the reference and methanol extract-treated groups (Figure 5D,F). Histopathological findings showed that the severity of lesions for the extracts was reduced in the *n*-hexane, ethyl acetate, aqueous, methanol, and reference groups, respectively (Figure 5A–F). 

The new endometrial gland formation (NEGF) seen in the ethyl acetate extract group showed that there were improvements in the endometrium and the formation of new endometrial tissue structure. In addition, the formation of new endometrial epithelial tissue (NEEF) in the endometrium indicated a remodeling state (Figure 5C).

Some degeneration was observed in collagen fiber (CF) structure and fibroblast cells (FC) in the connective tissue in the experimental groups. Increased collagen fiber (CF), new fibroblast cell formation (NFCF), and regular collagen fibers (RCF) were seen in the methanol extract- and buserelin acetate-treated groups, corroborating the regeneration of the endometrial tissue (Figure 5D,F).

## 3. Discussion 

Endometriosis is a gynecological disorder characterized by endometrial tissue implantation exterior to the uterus, mainly on the surface of the pelvis and ovaries, as well as in more distant tissues. It is a common gynecological condition, seen in 10% of women in the reproductive period and 2–50% of women diagnosed with infertility [2,5,30,31,32]. Symptoms such as headache, gastrointestinal disorders, dysmenorrhea, and painful sexual intercourse are observed and are characteristic of endometriosis [1,33]. While drugs used in the treatment of the disease provide relief of symptoms, they do not fully cure the disease, and often symptoms reoccur once the drugs are discontinued [34].

Due to the inadequacy of the effects of the existing drugs for the treatment of endometriosis and the high potential for side effects, the search for alternative treatments has increased [35,36,37]. There is a wealth of knowledge on plants used in the treatment of gynecological disorders in Turkey and many parts of the world [12,38,39,40,41,42,43,44]. In addition, many plants that can be used in the treatment of endometriosis have been reported in the scientific literature [45,46,47,48,49].

At the time of clinical appearance, most women already have established endometriosis, so it is difficult to provide experimental evidence for physiological roles in the pathogenesis of this disease in humans. Furthermore, ethical factors limit the performance of controlled experiments, and it is not possible to monitor the disease’s progress without performing continual laparoscopies. Therefore, research into the essential mechanisms by which menstrual endometrium adheres, invades, and establishes an efficient vasculature to persist in an ectopic site, along with the improvement of new therapeutic methods, is best implemented in experimental animal models. Even with the advantages of animal models, there are some limitations, such as small sample size, and the applicability of the results to humans.

Inflammatory cytokines play an important role in the peritoneal fluids of patients with endometriosis [50,51,52], with their levels increasing and being directly correlated with the stage of the disease [53,54]. TNF-α, a pro-inflammatory cytokine, induces cell differentiation, regeneration, and tissue restructuring. In addition, it increases peritoneal inflammation, by accelerating the passage of inflammatory factors into the peritoneal space, and enhances the permeability of localized blood vessels in the peritoneum [55]. The number of adhesive cells [56] and endometriotic stromal cells has been shown to increase [57] secondary to increased TNF-α levels. IL-6 is a cytokine that plays a role in the pathogenesis of endometriosis and is a biphasic immune molecule that acts as both a pro-inflammatory cytokine and an anti-inflammatory myokine [53]. IL-6 contributes to the implantation of endometrial cells [51], and its levels in the peritoneal fluid of women suffering from endometriosis are significantly increased [58]. VEGF, also known as a vascular permeability factor, is a heparin-linked glycoprotein that contributes to angiogenesis in endometriosis [58,59,60]. Levels of VEGF in the peritoneal fluid increase due to the prominent vascularization in the peritoneum in patients with endometriosis, and the level of VEGF increases to a greater extent in advanced stages of the disease [61,62]. Accordingly, there is a decrease in the amount of VEGF at the tissue level, consistent with our findings. In our study, the peritoneal TNF-α level of the methanolic extract-treated group decreased by 48.9%, the VEGF level by 50.8%, and the IL-6 level by 20.9% (Table 3). These results showed that *M. neglecta* led to a decrease in the increased cytokine levels associated with endometriosis. The decreased cytokine levels detected in the reference and methanol extract groups could probably cause the reduction of endometrial implants due to decreased inflammatory process and angiogenesis.

Polyphenols, consisting of six main groups of bioactive compounds, including flavonoids, lignans, non-phenolic metabolites, other polyphenols, phenolic acids, and stilbenes, are found in different proportions in plants. These polyphenols play an important role in endometriosis and are widely known for their multiple functional health-promoting benefits, especially antioxidant and anti-inflammatory properties. Due to their structurally different polyphenol compounds, these compounds are currently seen as promising candidates for developing new strategies for the treatment of endometriosis.

Many studies have shown the effects of polyphenolic compounds on estrogen receptors in endometriosis. For example, resveratrol has an estrogenic effect at low concentrations and an antiestrogenic effect at high concentrations and reduces cell proliferation, due to a decrease in epithelial estrogenic receptor α levels [63]. Curcumin reduces cell proliferation by decreasing estradiol levels [64]. Genistein has been reported to show antagonistic activity against endometriotic implants in the presence of estrogen in rats [65]. There are also studies showing that polyphenolic compounds, such as quercetin, apigenin, wogonin, rosmarinic acid, and curcumin, prevent the growth of endometrial tissue in various mechanisms [48,65,66,67,68,69]. Curcumin, quercetin, and resveratrol intervene in the regulation of cell proliferation and apoptosis, contributing to the decrement of ectopic tissue growth. While the effect of quercetin in signaling pathways is associated with cell proliferation [70], for curcumin, only the effect in cell cycle arrest has been described [64,71]. Intervention in apoptotic vias and regression of endometriotic implants has been demonstrated for these compounds [70,72,73,74,75,76].

Polyphenols highly inhibit the expression of IL-6 and TNF-α [31,53]. In studies on the effects of polyphenols on angiogenesis and VEGF levels, it has been determined that they inhibit angiogenesis in rats [6,8,77,78]. Recently, researchers have isolated several metabolites, especially quercetin from the *Malva* species, as having anti-inflammatory properties [79]. It has also been displayed in several experimental studies that *M. sylvestris* inhibits the production of prostaglandins D2, E2, and F2α [80], decreases TNF-α and IL-6 gene expression and interferes with the production of IL-1β and leukocyte migration in cell culture media [81]. Therefore, it appears that it can reduce the expression of pro-inflammatory factors and adhesion molecules with several anti-inflammatory components, which was also seen in our study. 

Various studies have shown that phenolic acids may be appropriate in the treatment of endometriosis, due to their antioxidant, anti-inflammatory, anti-tumor, and anti-angiogenic effects. In a study by Ferella et al., it was determined that rosmarinic acid, a phenolic acid, and carnosic acid, a phenolic diterpene, suppressed cell proliferation and reduced the size of endometriotic lesions in mice. In addition, it has been reported that rosmarinic acid promotes apoptosis in endometriotic tissue and inhibits oxidation, by reducing the accumulation of intracellular reactive oxygen species (ROS) in primary endometriotic stromal cells [67]. In our study, it was determined that the methanolic extract, prepared from the leaves of *M. neglecta*, was rich in phenolic acids. Especially malic, 4-hydroxy benzoic and salicylic acid were quite high, compared to other substances. These results showed that the phenolic compounds of *M. neglecta* may be responsible for the treatment of endometriosis.

## 4. Material and Methods

### 4.1. Plant Material

The above-ground parts of *M. neglecta* were collected from Amasya-Suluova, Turkey, in the first week of June 2019, and were dusted after drying in the shade. An herbarium sample was given to Gazi University, Faculty of Science, Department of Biology by Professor Dr. Hayri DUMAN (Herbarium no: GUEF3836).

### 4.2. Extraction Procedure

The powdered plant material was weighed, and 500 g was extracted with 5 L each of *n*-hexane, ethyl acetate (EtOAc), methanol (MeOH), and distilled water for 4 days, respectively. Then, it was filtered and the obtained extracts were combined and concentrated in a rotavapor at 40 °C under low pressure. The concentrated extracts were placed in a vacuum desiccator and allowed to dry completely. The yields of *n*-hexane, EtOAc, MeOH, and aqueous extracts were calculated as 6.78%, 8.96%, 14.83%, and 10.25%, respectively.

### 4.3. Determination of the Chemical Profile of Methanol Extract Prepared from Malva neglecta Leaves by HPLC

To determine the chemical profile of the methanolic extract prepared from *M. neglecta* leaves, the extract (100 mg) was re-dissolved in methanol (5 mL) and vortexed for 1 min. The extract was filtered into 2 mL HPLC bottles using a PFTE syringe filter (0.45 µm, Agilent Technologies, Darmstadt, Germany) for analysis. Then, *p*-hydroxybenzoic acid, hydroxytyrosol, caftaric acid, epicatechin, ferulic acid, 3-O-caffeoylquinic acid, quercetin-3-rutinosite, and quercetin-3-glucoside standards were dissolved in methanol and applied to the system.

The Agilent Technologies 1200 series system, coupled with a diode array detector, was used for the analysis of phenolic compounds. The chromatographic separation of the sample was performed using a mobile phase, consisting of solvent A (methanol:acetic acid:deionized water, 10:2:88, *v*/*v*/*v*) and solvent B (methanol:acetic acid:deionized water, 90:2:8, *v*/*v*/*v*) at a flow rate of 1 mL/min. The mobile phases were filtered through a 0.45 μm filter membrane and degassed in the sonicator for 30 min before analysis. A sample (10 μL) was injected onto a C18 column (Agilent-Zorbax Eclipse C18; Santa Clara, CA, USA) with a pore size of 3.5 μm, and maintained at a temperature of 25 °C for phenolic compounds analysis. Chromatograms were extracted at 320 nm, whereas absorption spectra were scanned in the range of 200–600 nm.

Peaks with purity greater than 95% were qualified. Identification of phenolic compounds was performed using the retention time and absorption spectra with available standard compounds in concurrent analyses. In the absence of standards, the definition was based on a comparison of absorption spectra reported in the literature [27,28,29]. The quantification of unknown compounds was based on calibration curves of those standard compounds having similar chromatographic response factors to the unknown compounds.

### 4.4. Bioactivity Studies

#### 4.4.1. Animals

Wistar albino female rats, aged 8–10 weeks, weighing 200–250 g, were obtained from the Experimental Animal Production and Research Laboratory of Kobay (Turkey), for the studies. All animals were hospitalized by the Guide for the Care and Use of Laboratory Animals, and the experiment was approved by the Experimental Animal Ethics Committee of Kobay (Protocol number: 233). For the animals to adapt to the environment, they were kept in laboratory conditions for at least three days before starting the experiment. During the waiting period, the animals were fed with water and standard pellet feed and housed in the laboratory at 21–24 °C at room temperature, 40–45% humidity and 12 h of light, and 12 h of darkness. Six animals in each group were used for the experiments.

#### 4.4.2. Preparation of Test Samples

In biological activity experimental models, the test samples were suspended in 0.5% sodium carboxymethyl cellulose (CMC) solution, with the help of an ultrasonic bath when necessary, and administered orally at a dose of 100 mg/kg via stomach gavage. The control group was given 0.5% CMC, which was used for the preparation of the test samples. In the endometriosis experimental model, test samples were administered to rats for 4 weeks.

Buserelin acetate was used as the reference material. During the experimental period, rats were administered subcutaneously once a week at a dose of 20 mg/kg.

#### 4.4.3. In Vivo Activity Assays

##### Endometriosis Rat Model

The rat model of endometriosis of Vernon and Wilson (1985) was applied [82].

Briefly, proestrus rats were anesthetized by intraperitoneal administration of 1 mL of ketamine hydrochloride (50 mg/mL) and 1 mL of xylazine hydrochloride (20 mg/mL). The abdomen of the rats was shaved and then disinfected with an iodine solution. After making a 3 cm incision using an abdominal scalpel, the abdomen was opened by separating the subcutaneous and muscle layers. The right uterus was removed and a 15 mm piece was taken. The piece was opened longitudinally and the endometrium layer was separated from the myometrium. The removed endometrial tissue piece was sutured to the abdominal wall of the same rat. The muscle layers of the abdomen were closed using silk thread. The second operation was performed 28 days after the first operation, and the endometriotic tissue regions and adhesions were evaluated. Endometrial implant dimensions were calculated by measuring height, width, and length with a micrometer.

The formula “π/6 × width × height × height” was used to calculate the ellipsoid volume [83].

Intra-abdominal adhesion scores were evaluated according to Blauer’s scoring system given below [84]:0:No adhesion1:Thin adhesion2:Thick adhesion in one area3:Spread of thick adhesion4:Adhesion including internal organs

After the abdomen was closed with the same procedure, the extracts prepared for the treatment group rats were applied to the control group rats with 0.5% CMC for 28 days. Buserelin acetate was administered subcutaneously to the reference group once a week. After the treatment, the rats were sacrificed, and the adhesions and endometriotic tissue sizes were re-evaluated and compared with the previous findings [48].

#### 4.4.4. In Vitro Activity Assays

##### Measurement of Cytokine Levels in the Peritoneal Fluid

In the endometriosis experimental model, TNF-α, (VEGF), and IL-6 levels were measured by taking peritoneal fluid twice, during the second operation and after the sacrifice [85].

##### Measurement of TNF-α, VEGF, and IL-6 and Levels in Peritoneal Fluid

TNF-α, VEGF, and IL-6 levels in the peritoneal fluid were measured using ELISA kits. Reagents, samples, and standards were prepared, based on the manufacturer’s instructions. An amount of 100 µL of standard and test samples were placed in each well of the 96-well plate and incubated at 37 °C for 2 h. Next, liquids were removed from each well, and 100 µL of biotin antibody was added to the wells and incubated at 37 °C for 1 h. After the incubation period, the wells were washed 3 times with a washing solution. After the washing process was completed, the wells were emptied and 100 µL of horseradish peroxidase (HRP)-avidin was added to each well and incubated again at 37 °C for 1 h. The wells were emptied and washed 5 times with a washing solution. Next, 90 µL of 3,3′,5,5′-tetramethylbenzidine (TMB) substrate was added and incubated for 30 min at 37 °C in the dark. In the last step, 50 µL of stop solution was added and reading was performed at 450 nm, using an ELISA microplate reader [85].

##### RNA Extraction

Total RNA was isolated from endometrium tissue using the RNeasy Mini Kit (Qiagen, Hilden, Germany), following the manufacturer’s protocol. Measurement of RNA concentrations was done using a NanoDrop ND-2000 spectrophotometer (NanoDrop Technologies, Wilmington, DE, USA). Total RNA was reverse-transcribed with a High-Capacity cDNA Reverse Transcription Kit (Invitrogen, Carlsbad, CA, USA), according to the manufacturer’s recommendations. 

##### The cDNA Synthesis and Quantitative Real-Time PCR (qRT-PCR)

For quantitative real-time PCR (qRT–PCR), 500 ng total RNA was reverse-transcribed using the High-Capacity cDNA Reverse Transcription Kit (Invitrogen). The resulting cDNA was analyzed by qRT–PCR, using the Luminaris Color HiGreen Low ROX qPCR Master Mix (2×) (Thermo Scientific, Waltham, MA, USA). The cDNA products were amplified using primers specific for VEGF (forward: 5′-CGGAAGATTAGGGAGTTT-3′and reverse: 5′-GGATGGGTTTGTCGTGTT-3′), TNF (forward: 5′-AAATGGGCTCCCTCTCATCAGTTC-3′ and reverse: 5′-TCTGCTTGGTGGTTTGCTACGAC-3′), and B-actin (forward: 5′-AGAGGGAAATCGTGCGTGAC-3′and reverse: 5′-CCATACCCAGGAAGGAAGGCT-3′), were used as a normalization control. All reactions were run on a Mic qPCR, (Bio Molecular Systems, Upper Coomera, QLD, Australia) with a 10-min hot start at 95 °C, followed by 40 cycles of a 3-step thermocycling programmer of the following three steps of denaturation, 15 s at 95 °C; annealing, 30 s at 60 °C; and extension, 30 s at 72 °C. The melt curve was at 60–95 °C in 0.5 °C intervals, and 2 s/step. Melting curves (60–95 °C) were examined to verify that a single product was amplified. All reactions were performed in duplicate, and relative expression levels were determined with the ΔCT method and reported as 2-ΔCt (fresh tissue samples) [86]. 

#### 4.4.5. Techniques for Histopathological Investigation

Firstly, all endometriotic tissues from the experimental groups were fixed with 10% formaldehyde. All tissues were detected using the Thermo Scientific Excelsior (ES) machine. The tissues were embedded in paraffin wax and blocks were prepared using the HistoCentre 2 machine. Subsequently, sections of 3.5 µm thickness were made from paraffin-embedded blocks using a Leica RM2255 microtome. The sections were stained with hematoxylin-eosin (HE) using the Shandon Varistan machine. Photographs of pathological endometriotic tissues were taken using Nikon Eclipse Ci with both polarizing attachment and a Digital Image analysis system, which were then examined under a light microscope. In the histopathological analysis, the severity of lesions in the implants was determined according to the presence of endometrial glands [87].

#### 4.4.6. Statistical Analysis

The GraphPad Prism 6.0 (San Diego, CA, USA) program was used for statistical analysis. An ANOVA test was performed on all parameters and then the Dunnett test was applied. The statistical significance of the experimental results, compared with the control and reference groups, was expressed in the following figures: *: *p* < 0.05; **: *p* < 0.01; ***: *p* < 0.001.

## 5. Conclusions 

In this study, the effects of *M. neglecta* leaves were investigated in a rat endometriosis model. Biological activity and phytochemical characterization established a high polyphenol content in a methanolic extract prepared from the leaves of *M. neglecta*. The extract showed high efficacy against endometriosis, likely due to the antiestrogenic activities of polyphenols, and their inhibitory activity on cytokines. The findings, taken together, suggest that the extract prepared from the leaves of *M. neglecta* may afford an alternative efficacious treatment for endometriosis. For the discovery of new and potent compounds for the treatment of endometriosis, further research on the isolation and identification of the active principle(s) of the MeOH extract is in progress. Moreover, there is no publication concerning the anti-endometriotic effect of *M. neglecta* in the literature. So, this effect of the plant was reported for the first time in the present study.

## Figures and Tables

**Figure 1 molecules-27-07869-f001:**
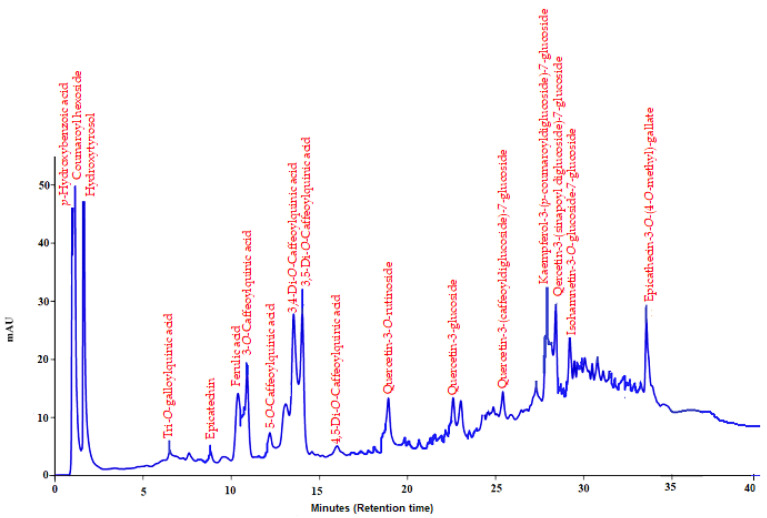
HPLC-DAD chromatogram of methanolic extract prepared from *M. neglecta* leaves.

**Figure 2 molecules-27-07869-f002:**
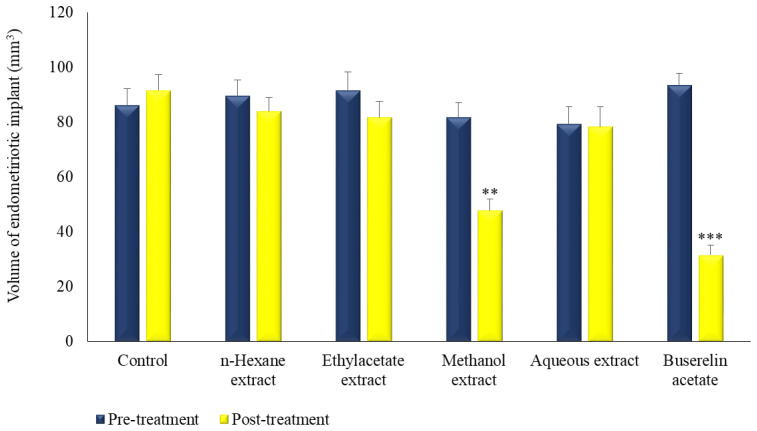
The effect of extracts prepared from the leaves of *M. neglecta* on endometriotic implant volumes in rats. ** *p* < 0.01, *** *p* < 0.001.

**Figure 3 molecules-27-07869-f003:**
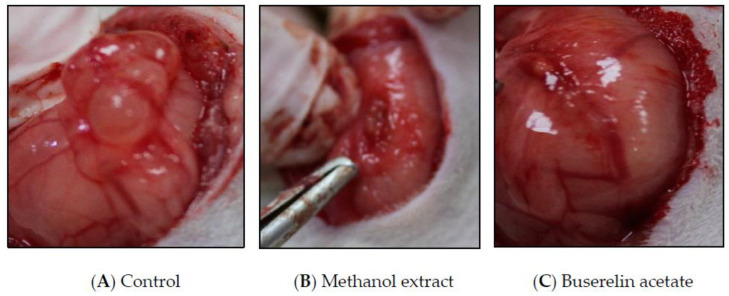
Post-treatment endometriotic implant images of control, methanol extract, and buserelin acetate.

**Figure 4 molecules-27-07869-f004:**
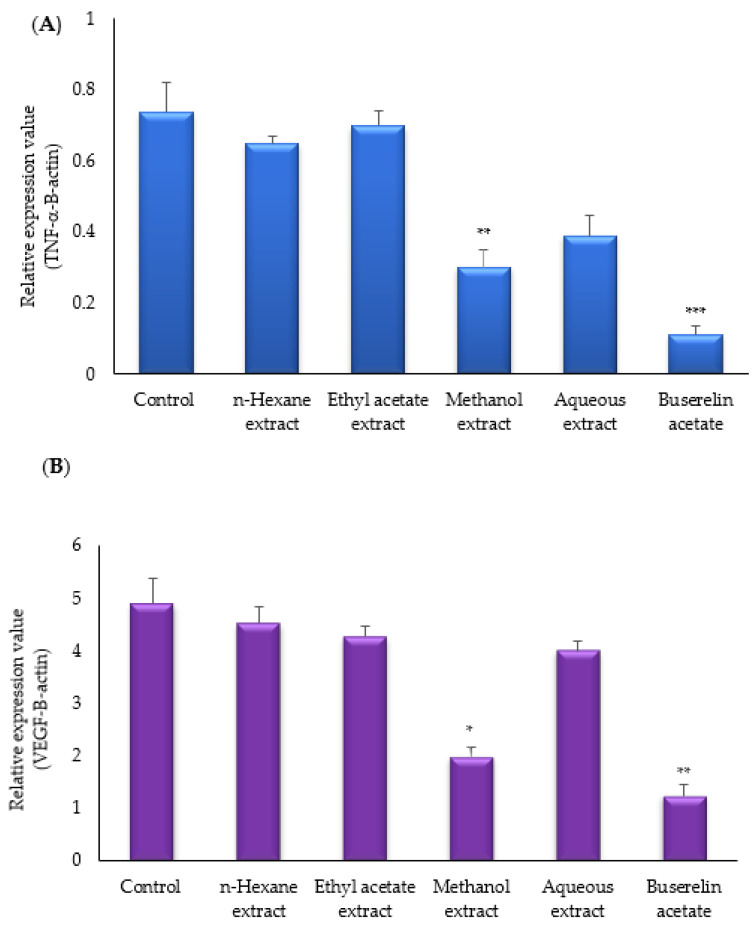
Effects of *M. neglecta* on mRNA (**A**) expression of TNF-alpha, (**B**) VEGF-A in endometrial tissue of rats. Values are means and Ds. Values were compared by ANOVA, * *p* < 0.05; ** *p* < 0.01; *** *p* < 0.001 vs. control.

**Figure 5 molecules-27-07869-f005:**
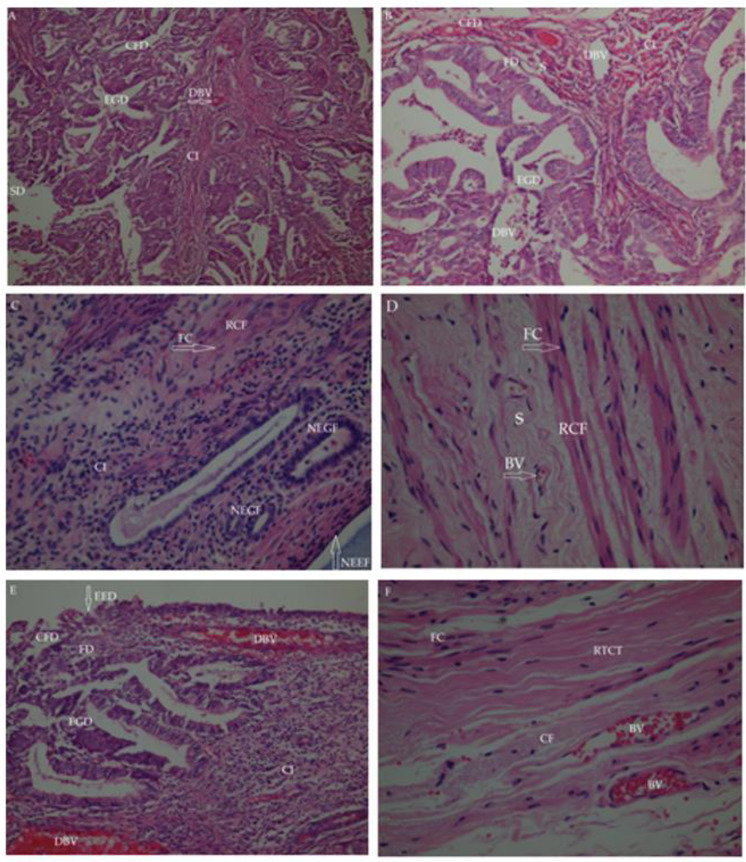
Histopathological findings of endometrial tissues with the treatment of test groups. (**A**) Control group; (**B**) The *n*-Hexane extract group; (**C**) Ethyl acetate extract group; (**D**) Methanol extract group; (**E**) Aqueous extract group; (**F**) Reference (Buserelin acetate) group. BV: Blood vessel; CFD: Collagen fiber degeneration; CI: Cell infiltration (inflammatory cells); DBV: Dilated blood vessel; EED: Endometrial epithelium destruction; EG: Endometrial gland destruction; FC: Fibroblast cell; FD: Fibroblast degenerative; NEEF: New endometrial epithelium formation; NEGF: New endometrial gland formation; RCF: Regular collagen fibers; RTCT: Regular tight connective tissue; S: Stroma; SD: Stromal destruction (connective tissue degeneration).

**Table 1 molecules-27-07869-t001:** Identification and composition of the polyphenolic profile of the methanol extract prepared from the leaves of *M. neglecta* using HPLC-DAD.

Retention Time (min)	Compound	Absorbance (nm)	Composition (mg/g)	Reference
0.9	p-Hydroxybenzoic acid	252	3.7 ± 0.3	[23]
1.0	Coumaroil hexoside	264	97.2 ± 3.1	[27]
1.7	Hydroxytyrosol	230, 280	107.1 ± 3.2	Standard
6.5	Tri-*O*-galloilquinic acid	270	1.6 ± 0.1	[28]
8.8	Epicatechin	230, 270	2.1 ± 0.2	Standard
10.3	Ferulic acid	327	5.3 ± 0.4	
10.8	3-*O*-Caffeoylquinic acid	305, 328	7.0 ± 0.5	[23]
12.1	5-*O*- Caffeoylquinic acid	305, 327	3.4 ± 0.2	
13.4	3,4-Di-*O*-caffeoylquinic acid	327	9.3 ± 0.6	[29]
14.0	3,5-Di-*O*-caffeoylquinic acid	328	7.4 ± 1.1	
16.1	4,5-Di-*O*-Caffeoylquinic acid	295, 327	1.3 ± 0.2	
18.8	Quercetin-3-*O*-rutinoside	256, 358	31.3 ± 1.1	Standard
22.7	Quercetin -3-glucoside	256, 352	24.2 ± 1.4	
25.3	Quercetin-3(caffeoyl diglucoside)-7-glucoside	270, 330	15.5 ± 0.6	[23]
28.0	Kaempferol-3-(*p*-kumaroil diglukozit)-7-glukozit	245, 330	37.1 ± 3.3	[23]
28.3	Quercetin-3-(sinapoyl diglucoside)-7-glucoside	250, 340	27.1 ± 1.4	
29.1	Isorhamnetin-3-*O*-glucoside-7-glucoside	253, 340	21.1 ± 1.1	[28]
33.7	Epikateşin-3-*O*-(4-*O*-metil)-gallat	275	31.4 ± 3.5	
**Total amount**	**433.1**			

**Table 2 molecules-27-07869-t002:** Intra-abdominal adhesion scores of extracts prepared from the leaves of *M. neglecta* on endometriotic implants in rats.

Group	Extract	Adhesion Scores ± S.E.M.		Decrease in Adhesion Scores (%)
		Pre-Treatment	Post-Treatment	
**Control**		3.4 ± 0.7	3.5 ± 0.9	-
** *M. neglecta* **	*n*-Hexane	3.2 ± 0.5	2.9 ± 0.8	9.4
	Ethyl acetate	3.3 ± 0.9	2.7 ± 0.6	18.2
	Methanol	3.3 ± 0.8	**1.4 ± 0.5 ***	57.6
	Aqueous	3.1 ± 1.1	2.9 ± 0.9	6.5
**Reference**	Buserelin acetate	3.3 ± 0.9	**0.0 ± 0.0 *****	100

* *p* < 0.05; *** *p* < 0.001; S.E.M.: Standard Error of Mean; Bold and asterisk indicate statistical significance.

**Table 3 molecules-27-07869-t003:** The effect of *M. neglecta* on cytokine levels in peritoneal fluids in a rat endometriosis model.

Group	Extract	Peritoneal TNF-α Level (pg/mL) ± S.E.M.		Peritoneal VEGF Level (pg/mL) ± S.E.M.		Peritoneal IL-6 Level (pg/mL) ± S.E.M.	
		Pre-Treatment	Post-Treatment	Pre-Treatment	Post-Treatment	Pre-Treatment	Post-Treatment
**Control**		9.3 ± 3.1	8.8 ± 2.0	25.1 ± 5.2	27.4 ± 4.8	51.5 ± 9.2	52.5 ± 7.1
** *M. neglecta* **	*n*-Hexane	9.1 ± 2.8	8.6 ± 2.1	19.1 ± 4.6	17.3 ± 5.1	48.7 ± 7.3	46.6 ± 6.2
	Ethyl acetate	9.9 ± 3.2	8.3 ± 2.2	23.3 ± 5.8	20.2 ± 7.9	50.4 ± 5.4	57.9 ± 4.8
	Methanol	9.6 ± 2.4	**4.9 ± 0.7 ***	24.6 ± 5.1	**12.1 ± 3.8 ***	43.1 ± 5.1	**34.1 ± 4.6 ***
	Aqueous	8.4± 3.3	7.9 ± 2.6	18.7 ± 4.8	19.0 ± 4.1	50.4 ± 6.8	48.2 ± 7.3
**Reference**	Buserelin acetate	8.8. ± 1.4	**2.3± 0.7 *****	20.4 ± 5.1	**9.2 ± 2.8 *****	49.2 ± 4.9	**21.2 ± 4.3 *****

*: *p* < 0.05; ***: *p* < 0.001; S.E.M.: Standard Error of Mean; Bold and asterisk indicate statistical significance

## Data Availability

Not applicable.

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
