# Peer review of "The Phytochemical Profile and Biological Activity of Malva neglecta Wallr. in Surgically Induced Endometriosis Model in Rats"

_molecules, 2022, doi:10.3390/molecules27227869_

Round 1
Reviewer 1 Report
Due to the fact that the effectiveness of modern drugs for endometriosis is insufficient and the disease frequently recurs and the potential side effects are high, researchers are trying to find alternative treatments. Medicinal plants have been used traditionally for centuries as potential healing agents. Therefore, in this work, the authors undertook research on the influence of M. neglecta leaves on endometriosis. The choice of this plant was dictated by the fact that it was used in the treatment of female infertility. The plant material was extracted successively with n-hexane, ethyl acetate and methanol.
Experimental endometriosis was surgically induced in six-week-old female non-pregnant Wistar albino rats by autotransplantation of endometrial tissue into the abdominal wall.
Both the endometrial volume and the adhesion score decreased significantly in the group treated with the methanol extract. In addition, significant decreases in TNF-α, VEGF and IL-6 levels were observed in animals treated with methanol extract.
HPLC results indicated that the methanol extract prepared from M. neglecta leaves is rich in phenolic acids. Especially malic, 4-hydroxybenzoic and salicylic acid are quite high compared to other substances. These results showed that phenolic compounds may be responsible for the treatment of endometriosis.
This work provides important information on the therapeutic use of M. neglecta and may be a promising alternative in the treatment of endometriosis.
In the Discussion section, the authors should emphasize the advantages of their findings compared to the results presented by other scientists.
It may be released after some minor revision.
Author Response
In the discussion section, the advantages of our findings are compared to the results presented by other scientists.
Reviewer 2 Report
1. The English need improvement since there are some grammatical and syntax errors in the manuscript. For example, the words “a reference groups” may be as “a reference group of reference groups”; “was done” as “were done”; “and seen” as “and is seen”; “a porridge” as “porridge”; “with HPLC” as “of HPLC”; “fibers (RCF) was” as “fibers (RCF) were”; “with treatment” as “with the treatment”; “treatment has” as “treatments have”; “as vascular” as “as a vascular”; “and IL-6” as “and the IL-6”; “20.9% was” as “20.9%”; “of chemical” as “of the chemical”; “with iodine” as “with an iodine”; “washing” as “a washing”; “a reading” as “reading”. The grammar mistakes which are not mentioned here are also to be checked and corrected properly.
2. There are some typing mistakes as well, and authors are advised to carefully proof-read the text. For example, the words “extract treated” may be as “extract-treated”; “ethylacetate” as “ethyl acetate”; “qualtified” as “qualified or quantified”. The typos not mentioned here are also to be checked and corrected properly.
3. Check the abbreviations throughout the manuscript and introduce the abbreviation when the full word appears the first time in the abstract and the remaining for the text and then use only the abbreviation (For example, TNF-α, VEGF, IL-6, PCR, etc.,). Make a word abbreviated in the article that is repeated at least three times in the text, not all words need to be abbreviated. The use of abbreviations in the absraact may distract readers who wish to quickly skim through several publications before deciding to read one in full. It may therefore help to write out terms fully in this section.
4. The full form of the species should be given when the first time appears in both the abstract and in the remaining part of the manuscript and it should be followed by only the first letter of the genus (e.g., Malva neglecta when the first time appears and followed by M. neglecta). The plant name should be italic all over the manuscript.
5. The authors have used only four keywords in the manuscript. Additional keywords may be included and the keywords that are not in the title. The keywords should assist computer searches to find your specific article.
6. The introduction part appears less informative about the endometriosis, thus this section should be indicated as detailed to understand the present investigation in clear. For example, the prevalence, risk factors or pathophysiology etc.
7. Scientific evidence should be supported with proper references of the medicinal benefits of the plant chosen, since it is not given anywhere in the introduction section..
8. The table and figure legends should be improved and a proper footnote should be given. All legends should have enough description for a reader to understand the table and figure without having to refer back to the main text of the manuscript. For example, the necessary expansion may be given for abbreviations use (TNF-α, VEGF, IL-6).
9. In materials and methods, the author should include the age group of the animal model used in the present findings.
10. In the materials and methods, the authors may cite references for standard protocol, instead of mentioning kid or manufacture instructions, if reference is given with it and the same should be added in the reference section. For example, measurement of TNF-α, VEGF, IL-6 is not supported with proper references.
11. The conclusion section appears to be just a detailed summary of results/observations. All conclusions must be convincing statements on what was found to be novel, impactbased on the strong support of the data/results/discussion. Moreover, the authors may be included the limitation of the present findings and future direction for a better understanding of the manuscript.
12. The reference are not cited properly it should be checked and corrected properly. For example, few reference full name of the journals has been given and for others only short form has been given. It should be carefully checked and corrected as per the journal format.
Author Response
- The grammar mistakes were checked and corrected properly.
- Typing mistakes were checked and corrected properly.
- Abbreviations throughout the manuscript were checked and introduced the abbreviation when the full word appears the first time in the abstract and the remaining for the text and then used only the abbreviation.
- The full form of the species was given when the first time appears in the abstract, the remaining part of the manuscript, and the subtitle. The plant name was written in italic all over the manuscript.
- Some keywords were added.
- The prevalence, risk factors or pathophysiology of endometriosis were added.
- References were added.
-
Results were mentioned above tables and figures. Footnotes are not added under Tables and Figures to avoid repetition of the same information.
- The age group of the animal was added.
- Done.
- Done.
- References were checked and mistakes were revised.
Reviewer 3 Report
This manuscript tries to evident the ameliorating effects of the phytochemical contents of Malvaneglecta Wallr. on a mouse endometriosis model in which experimental endometriosis was surgically induced in six-week-old female, non-pregnant, Wistar albino rats by autotransplant of endometrial tissue to the abdominal wall. The authors found that both endometrial volume and adhesion score decreased significantly in the group treated with methanol extracts. Moreover, TNF-α, VEGF, and IL-6 levels in animals administered the methanol extracts were significantly decreased. HPLC results showed that the activities caused by the methanol extract of M. neglecta were due to the polyphenols.
Overall, the results were clearly presented, suggesting that the compounds in the methanol extract of M. neglecta may clinically be applied to human patients of endometriosis in the future. However, the results represented only the net values of all the 18 polyphenol compounds contained in the methanol extract of M. neglecta. It is likely that some compounds have negative effects on endometriosis. The patients would have benefited more if the authors can differentiate and identify specific compounds that are positive and negative for endometriosis. Alternatively, they may look into the literature and choose to test the most possible compound responsible for the ameliorating role of the methanol extract of M. neglecta.
Author Response
Unfortunately, in this study, it was not possible to determine the only substance/s responsible for the effect, since the study material could not be obtained in the amount to be administered to the experimental animals. However, in future studies, it was aimed to determine the substance/s that may be responsible for the effect and to investigate the effect by purchasing them commercially.
Round 2
Reviewer 2 Report
1. The abbreviation should be avoided in the keywords (PCR) and give only full form in the keywords.
Author Response
The abbreviation (PCR) was deleted and the full form was written in the keywords.
Reviewer 3 Report
Suppose it's impossible for the authors for now to identify specific phytochemicals responsible for the effects of the methanol extract of M. neglecta. In that case, they should at least navigate the bulk literature and try to rationalize and predict in the Discussion which chemical compounds likely contribute to the ameliorating effects of the methanol extract on endometriosis. The efforts may enrich the scientific soundness and biological significance.
Author Response
Literature information on chemical compounds possibly contributing to the curative effects of methanol extract on endometriosis was added to the discussion.